



# RASCAL v1.0.0: An Open Source Tool for Climatological Time Series Reconstruction and Extension

Álvaro González-Cervera[1, 2] and Luis Durán[2]

[1]interMET Sistemas SME, Madrid, Spain
[2]D. Física de la Tierra y Astrofisica, Facultad de Física, Universidad Complutense de Madrid, P. Ciencias s/n, Madrid, 28040, Madrid, Spain

**Correspondence:** Álvaro González-Cervera (alvaro@intermet.es)

**Abstract.**

The reduction of in situ observations in recent decades poses a potential risk of losing crucial information in regions where local effects significantly shape their climatology. Reanalyses face challenges in examining climatologies with highly localized effects, particularly in regions with intricate orography. Empirical downscaling methods offer a cost-effective and easier to implement in new areas alternative to dynamic downscaling methods. This article introduces RASCAL, an open-source Python tool designed to address gaps in observational climate data, especially in regions with limited long-term data and significant local effects, such as mountainous areas. Employing an object-oriented programming style, RASCAL's methodology effectively links large-scale circulation patterns with local atmospheric features, using the analog method in combination with principal components analysis (PCA), outperforming reanalysis in conveying climatic characteristics. The package contains routines for preprocessing observations and reanalysis data, generating reconstructions using various methods, and evaluating the reconstruction's performance in reproducing the time series of observations, statistical properties, and relevant climatic indices. Its high modularity and flexibility allows fast and reproducible downscaling. The evaluations carried out in central Spain, near a mountainous area and an urbanized area, demonstrate that RASCAL performs better than the ERA20C and ERA20CM reanalysis in terms of $R^2$, standard deviation, and bias. This is particularly evident in the reconstruction of monthly total precipitation. It is worth noting that RASCAL generates series with statistical properties, such as seasonality and daily distributions, that closely resemble observations, thus addressing the limitations of reanalysis biases. This addresses the limitations of reanalysis biases and confirms the potential of this method for conducting robust climate research. The adaptability of RASCAL to diverse scientific objectives is also highlighted. However, there are challenges to consider, such as the requirement for long-term data series and susceptibility to disruptions caused by changes in land use or urbanization processes. Despite these limitations, RASCAL's positive outcomes offer opportunities for comprehensive climate variability analyses and potential applications in downscaling short-term forecasts, seasonal predictions, and climate change scenarios. The Python code and the Jupyter Notebook for the reconstruction validation are publicly available as an open project.



## 1  Introduction

The origins of meteorological observation can be traced back to ancient civilizations, where people began to notice patterns in the weather and celestial phenomena. However, it wasn't until the 17th century that systematic weather observations began in earnest with the development of instruments such as the mercury barometer and the thermometer by scientists such as Evangelista Torricelli and Daniel Gabriel Fahrenheit (Barry and Chorley, 2009). An early example of this interest in observing the atmosphere using instruments is the Central England Temperature record (CET) (Manley, 1974) which is one of the longest

instrumental temperature records in the world, dating back to 1659. It provides a continuous monthly temperature series for the central England region and is often used as a proxy for temperature variations in Western Europe. Other examples of early weather monitoring date back to the 18th century, such as the Paris-Montsouris observations in France (Moisselin et al., 2002), the Zentralanstalt für Meteorologie und Geodynamik in Austria (Vienna) (Auer et al., 2007), the Uppsala University observations in Sweden (Bergström and Moberg, 2002), or the earliest observations recorded in Iberian Peninsula like those

starting in Seville (Spain) in 1780 (Domínguez-Castro et al., 2014). Since these first observations begun, the number of surface meteorological observatories worldwide has increased significantly, as shown in Fig. 1a.

The critical role played by surface meteorological stations in climate monitoring and research is emphasized by the Intergovernmental Panel on Climate Change (IPCC) in all its assessments and reports (IPCC, 2021). One of the aspects addressed is the need to maintain high quality and consistent data following high standards of quality assurance and control (Begert et al.,

2005). These kind of procedures are essential to ensure that the data collected are homogeneous, accurate and reliable. Errors or inconsistencies in the data can lead to erroneous climate assessments and predictions (Yang et al., 2005). Another important fact mentioned is the need for a dense network of surface meteorological stations around the globe to provide comprehensive coverage of different regions and climates. Dense monitoring networks are less common in remote or less densely populated regions or where the environmental conditions are too harsh to operate and maintain the instruments (Dinku, 2019; Fan et al.,

2020; Schween et al., 2020).

It has been commonly accepted that surface meteorological stations are still the best way to identify long-term trends and variability in climate. They have also proven to be critical for validating and calibrating other atmospheric databases such as those obtained from satellites or remote sensing instruments (Salio et al., 2015; Emery et al., 2001; Huang et al., 2019). They are also a key element for the development and validation of gridded databases obtained by numerical models such as

reanalyses (Molina et al., 2021; Bell et al., 2021; Lavers et al., 2022; Bonshoms et al., 2022). More recently, meteorological measurements have become an essential element of machine learning methods applied to atmospheric modeling. These methods implicitly account for all the involved physics through complex mathematical interrelationships of empirical nature, which require historical meteorological observations for training (Appelhans et al., 2015; Sachindra et al., 2018; Lam et al., 2022).



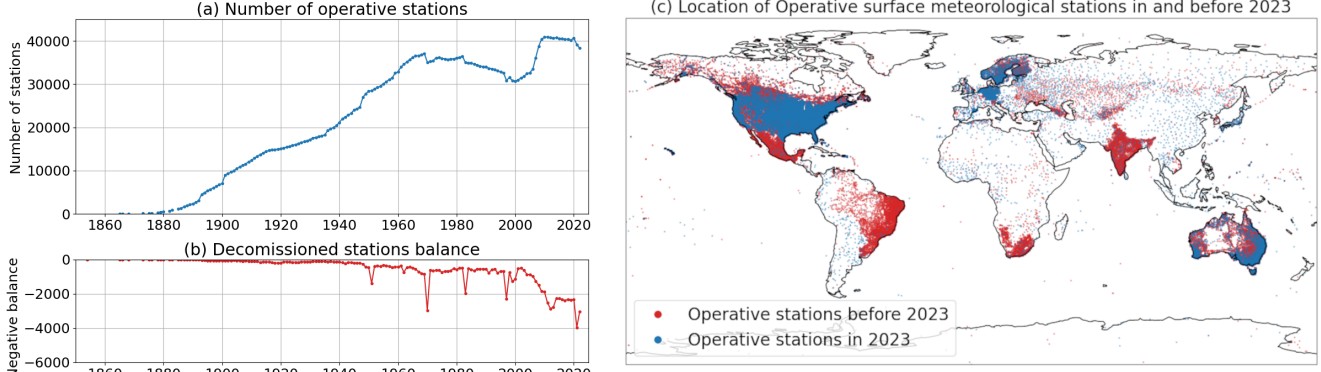

**Figure 1. (a)** Total number of operative stations from 1850 to 2023. **(b)** Balance of decommissioned stations in the same period. The negative value means the stations were no longer operative, and its absolute value represents the number of decommissioned stations. **(c)** Localization of all stations from 1850 to 2023, with operative stations in 2023 marked in blue and decommissioned stations until 2023 marked in red. Data obtained from the Global Historical Climatology Network daily (GHCNd, https://www.ncei.noaa.gov/products/land-based-station/global-historical-climatology-network-daily, accesed on 15 November 2023)

Due to the important role played by surface observations in climate assessing and weather forecasting, several countries established and expanded their surface meteorological observatories during the 20th century, trying to cover as much territory as possible (Klein Tank et al., 2002). From a few hundreds of surface stations at the end of the nineteenth century to several thousand at the end of the twentieth century (Fig. 1a). However, the results are very uneven around the world (Fig. 1c), with important areas of the world still under monitored.

After the world's first weather satellite was launched in 1960 (TIROS-1), satellite weather observations became common and began to offer multiple advantages over on-site weather observations (Purdom and Menzel, 1996). For instance, they allow for global coverage and cost-effectiveness since they do not require an extensive network of ground-based weather stations to cover vast areas. Even sometimes, satellite observations have shown to outperform surface measurements (Pinker et al., 2005; Heft-Neal et al., 2017). But satellite weather observations also have limitations. For example, they have difficulties accurately measuring conditions at the Earth's surface, their data availability is highly dependent on cloud cover, they often exhibit long-term instrument drift, and they have calibration issues in remote areas where surface observations are unavailable. Currently, satellite measurements are crucial for assessing the Earth's atmospheric conditions and perform the numerical weather prediction (Rabier, 2005). Despite the great improvement achieved in numerical weather prediction thanks to satellites, they may be behind the gradual decrease in the number of operational surface meteorological stations around the world, as illustrated in Fig. 1b.

Another potential factor contributing to the decline in the number of active surface weather stations worldwide in recent decades is the use of model reanalyzed data to conduct climate research (Dee et al., 2014; Hersbach, 2016). Model reanalyses use a combination of observational data sources, such as in-situ surface weather observations, satellite data, and others, to



generate a gridded and consistent dataset of weather and climate information from the past. The resulting data sets are comprehensive, homogeneous and have strong climatological consistency. They cover global areas, enabling analysis where in-situ
data is not accessible. They are certainly useful for studying broad climate patterns and long-term climate trends, and could theoretically be used to fill gaps or extend the temporal and spatial coverage of observations. However, they suffer significant losses with regards to temporal and spatial resolution as well as information relating to local phenomena. Global reanalyses have inherent difficulties to provide fine-scale details that are often missed in the physics of the models or are meaningless at the low resolution considered (Bromwich et al., 2007; Kaiser-Weiss et al., 2015; Gleixner et al., 2020).

This study is organized as follows: Sect. 2 discusses the motivation behind developing code to generate time series that capture climatological local phenomena. Sect. 3 provides a detailed description of the implemented method, while Sect. 4 describes the model structure and implementation. In Sect. 5, we evaluate the performance of the package by downscaling the daily maximum and minimum temperatures and precipitation of four stations near a mountainous region in central Spain. We draw final conclusions and important remarks in Sect. 6.

**2   Motivation**

Several factors, such as the use of satellite and reanalysis data for climate studies, along with other complex socio-political and economic events, may be behind the steady and heterogeneous decrease observed in the number of surface stations in operation worldwide during the last decades (Fig. 1). Although it may seem that global weather data is fully addressed through the more precise reanalyses available nowadays, there may be a hidden loss of information about local phenomena that only surface
weather stations are able to capture. When historical meteorological data is not continued indefinitely or if interrupted, many the resources invested during decades are lost. In addition to these interrupted time series, there are also numerous surface meteorological observation series of good quality around the world that are the result of short-term campaigns or very recent initiatives. These time series also provide a wealth of information on local processes, but their short duration is still insufficient for climatological analyses (Durán et al., 2017).

On the other hand, we have global reanalyses that span the entire 20th century and part of the 21st century, which provide a valuable alternative, but their low resolution limits the phenomena under analysis (Poli et al., 2016). Downscaling of reanalysis data in order to obtain pseudo-observations or finer gridded meteorological fields has been performed since the inception of reanalysis and can provide the best of both worlds. Two general approaches to downscaling are: physical downscaling and empirical downscaling, with statistical downscaling being a subset of the latter when statistical methods are used. Physical
downscaling is achieved by using higher resolution physical models that account for lower scale phenomena nested within the reanalysis fields (Lo et al., 2008; Durán and Barstad, 2018; Wang et al., 2021). On the other hand, empirical downscaling relies on observational data to establish empirical relationships between the large scale fields provided by the reanalysis and the local phenomena seen in the observations (Wilks, 2011; Bürger, 1996; Boé et al., 2006).

Numerous papers scrutinize the advantages and disadvantages of the various methods (Hewitson and Crane, 1996; Hanssen-
105 Bauer et al., 2003; De Rooy and Kok, 2004). As a rule of thumb, empirical techniques are generally less computationally



intensive than physical downscaling and may yield better results at a lower cost. However, empirical downscaling is only feasible when a long and uniformly collected dataset of observations is available. Assuming the hypothesis that there is a connection between the large-scale phenomena shown by the reanalysis and the local phenomena captured by the observations is also essential. In the context of downscaling climate change scenarios, this issue is subject to debate unless we consider a
second hypothesis that suggests a shift in the relative frequency of past demonstrated forcings without any inherent change in the phenomenological relationship between scales.

Regardless of the chosen regionalization method, combining reanalysis and surface observations to create long and homogeneous time series requires a significant amount of effort. Setting up a dynamic regionalization system can be expensive in terms of both computation and human resources, but even a relatively simple statistically-based regionalization method entails
a learning curve that may discourage or slow down certain climate studies. This work introduces and explains RASCAL v1.0.0 (Reconstruction by AnalogS of ClimatologicAL time series), an open-source tool for climatological time series reconstruction and extension using statistical downscaling. The primary goal of RASCAL is to promote and accelerate rigorous climate research in regions where surface meteorological observations are insufficient for climate analysis and where relevant regional and local meteorological processes can only be captured through in-situ observations. RASCAL could prove highly beneficial
for mountain climate research and other areas with unique microclimates, such as river valleys, forests, caves, or canyons.

## 3 Methods

RASCAL is based on the analog method, a widely used empirical downscaling technique in climate research (Zorita and Von Storch, 1999). It is based on the premise that large-scale atmospheric conditions tend to produce comparable local weather patterns, allowing the prediction of local conditions for a day without real-time observations. This is done by identifying an
analog day from General Circulation Models (GCMs), such as reanalyses, and assigning its local conditions. This approach allows the study of climate variability over an extended time frame, providing valuable perspectives on long-term patterns and connections between different geographic locations, while also incorporating important local factors into the analysis. (Hidalgo et al., 2008; Benestad, 2010; Abatzoglou and Brown, 2012; Saavedra-Moreno et al., 2015; Shulgina et al., 2023)

### 3.1 The analog method for time series reconstruction

The analog method is nonlinear technique that relies on the identification of strong statistical relationships between two fields: the predictor variable extracted from GCM products, and the predictand variable obtained from local historical observations. To predict an atmospheric feature (the predictand) for a given day, this method searches for the day with the most similar predictor field in the historical record and uses its atmospheric features to make a prediction, allowing the reconstruction of missing data points (Lorenz, 1969; Horton et al., 2017).

To incorporate the relationship between large-scale meteorological patterns and local weather, the analog method is often combined with Principal Component Analysis (PCA). The PCA reduces the high dimensionality of the atmospheric phase space by generating an orthogonal basis of vectors that represent the main directions of variability. As a result, only a limited





set of coefficients, called principal components (PCs), are required to represent the atmospheric state (Wilks, 2011). The resulting values of the PCs are considered the predictand. To identify the best analog, a pool of the N closest neighbors in the

PC space is constructed, using the Euclidean distance as the criterion for determining closeness.

Various similarity methods can be used to select the best analog or group of analogs from the pool. The most straightforward method is to choose the closest day in the PC space. However, similar synoptic patterns can sometimes produce different local weather if the role of other variables or more complex phenomena is not taken into account. Therefore, assigning the day with the most similar predictor field pattern as the analog day may not always be accurate. To avoid making the reconstruction

method too complex, one solution is to select the N closest days from the pool of analogs instead of just one. Then, perform a weighted average by the square of the distance in space of the PCs of those days. This way, the days with a more similar synoptic pattern have more presence in the average, while also considering possible phenomena that have not occurred on the closest day but on the other similar days.

Averaging can impact the distribution of the variable by smoothing the data and removing extreme values. To preserve

extremes while still accounting for possible phenomena beyond the similarity of synoptic patterns, bias reduction methods such as quantile-mapping can be used. This technique employs a 'mapping variable' and examines the quantile of the day to reconstruct the distribution of this variable in the analog pool. The method first examines the distribution of the predictand variable in the analog pool and selects the day that occupies the same quantile as the best analog. This approach improves the representation of extreme events. However, it is crucial to note that the mapping variable must have a strong correlation with

the predictand variable. One possible solution is to use the reanalysis predictand variable as the mapping variable.

The analog method fundamentally involves the reorganization of observed time-series data, aiming to maintain the statistical characteristics of the original dataset. The efficacy of this method relies on ensuring that the downscaling and training periods exhibit a comparable climatic context (Zorita et al., 1995). The reconstruction capability of the analog method is constrained by the temporal extent and accuracy of historical observations. This means that it cannot replicate unobserved events and,

therefore, cannot reproduce new record values in the context of climate change. However, its utility becomes pronounced in scenarios where external climate forcing induces shifts in the frequency of observed phenomena. In essence, it serves as a valuable tool for discerning alterations in the occurrence patterns of documented events.

## 4   Model Structure

The empirical downscaling techniques involve laborious steps that must be carefully addressed to ensure the quality of the local

climate series reconstructions, as pointed out in Boateng and Mutz (2023). RASCAL is a Python library that implements the analog method in a clear and simple way. It is an object-oriented library with four main blocks or classes: *Station*, *Predictor*, *Analogs* and *RSkill*. This library is a valuable complement to other empirical downscaling libraries, such as pyESD from Boateng and Mutz (2023), which is based on machine learning downscaling methods and focus on generating monthly time series. RASCAL is based on classical statistical methods, which produce results that are easier to interpret physically, and

additionally, it is more focused on daily resolution reconstructions rather than monthly, which allows for the calculation of





relevant daily climate indices. This section describes these components and their implementation workflow, with examples of code used for the reconstruction of daily total precipitation.

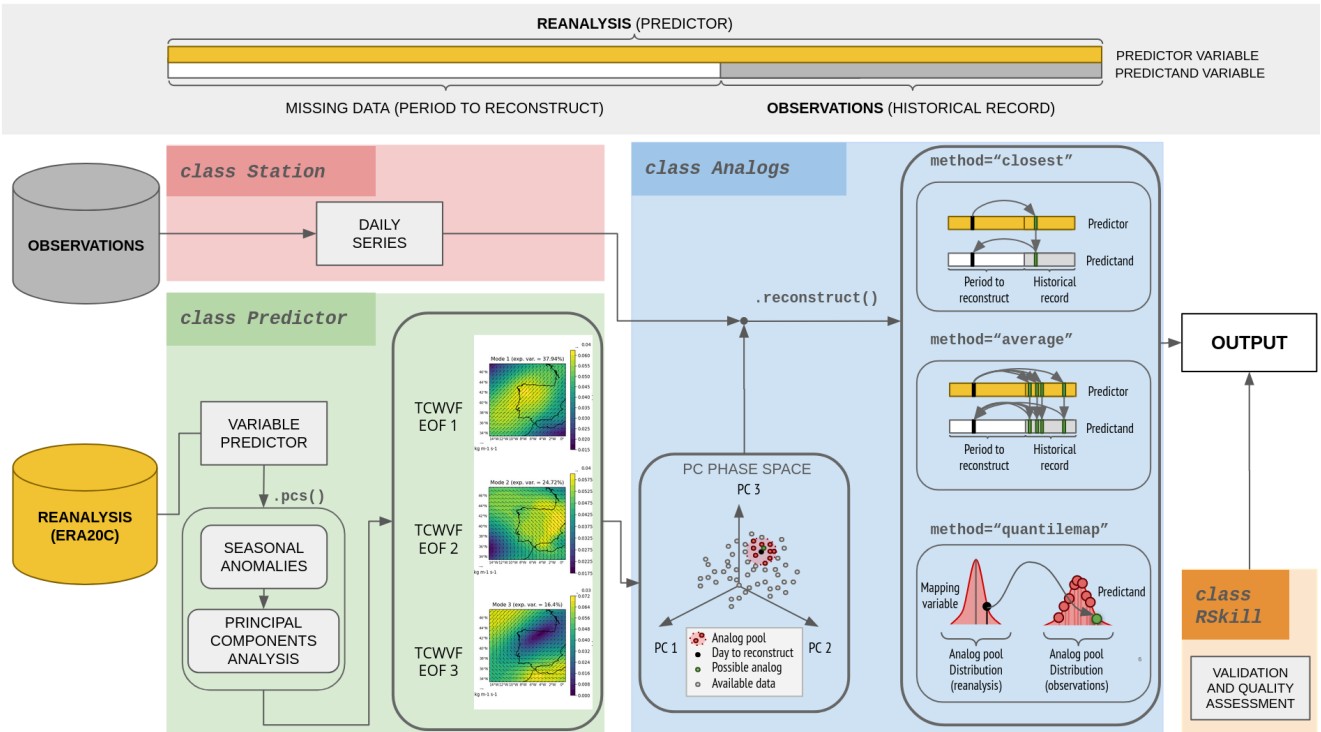

**Figure 2.** RASCAL main features and workflow. The colored boxes highlight the principal classes, and within them, the featured methods and objects are shown. An example of the EOFs obtained for the Total Column of Water Vapor Flux (TCWVF), used as a precipitation predictor, is included in the Predictor class box.

## 4.1 Station class

The analog method requires (1) homogeneous time series of observational data, and (2) a reanalysis dataset or GCM product

that covers both the period to be reconstructed and the period of historical observations. The *Station* class retains the information about the historical record, including metadata about the observation point such as its name, elevation, altitude, and longitude, as well as the observational data of the variable to be predicted. The historical record must have a daily to sub-daily resolution, and it is assumed to be homogeneous. The data is preprocessed to extract the desired meteorological variable, such as maximum, minimum, mean, or total accumulated, in the form of selected daily quantities. The code below provides an

example of how to extract the historical time series. The path should include a CSV file with the variable name and a meta.csv file with the metadata.

```
station = Station(path="./data/observations/")
```





```
station_data = station.get_data(variable="PCP")
```

## 4.2 Predictor class

The analog method has the benefit of low subjectivity due to its limited parameters for adjustment (Wetterhall et al., 2005). However, selecting and processing the predictor correctly is crucial for achieving accurate local weather reproduction. The selection should be based on our knowledge of atmospheric dynamics and the local climate of the study area, as pointed out by other authors (Boateng and Mutz, 2023). After selecting a predictor variable that is expected to have a strong relationship to the predictand variable, it is necessary to choose a predictor field domain that can identify relevant synoptic patterns for the study

area. These fields should be carefully grouped for each day. Although the analog method is based on recognizing patterns in a single predictor field, it is possible to use multiple variables within the same field. To use vector fields with multiple components or include different variables, it is necessary to construct a composite field by concatenating each variable on the longitude axis. This results in a single field with dimensions of (time, latitude, number of components x longitude).

These steps are implemented in the class *Predictor*. It takes as input the paths to the files of the chosen predictor, and allows

to select the limits of the field domain and to group them for each day, taking only one hour, or computing the sum or the average of all the available hours. The composite field is obtained when the *mosaic* option is set to *True* and more than one different variable is detected within the files of the input paths. The code example below selects both horizontal components of the Total Column of Water Vapor Flux (TCWVF) as the precipitation predictor from 1900 to 2010, and uses the field at 12:00 of each day in a limited domain.

```
predictor_files = rascal.utils.get_files(
    nwp_path="./data/reanalysis/",
    variables=["SURF_71.162", "SURF_72.162"],
    dates=["1900", "1901", ..., "2010"],
file_format=".grib"
)

predictors = Predictor(
    paths=predictor_files,
grouping="12hour_1D_mean",
    lat_min=20,
    lat_max=80,
    lon_min=-60,
    lon_max=20,
mosaic=True
)
```





Once the predictor field is chosen, the PCA is performed. The PCA is implemented as the method *Predictor.pcs()*. To perform the PCA is necessary to calculate the anomalies of the predictor field. In this method is possible to choose the months of each season and the number of seasons. The the number of principal components to be used, and the scaling of the PCs. This scaling will subsequently influence the selection of a pool of the N closest days. This method wraps the phyton library eofs (Dawson, 2016) using xarray (Hoyer et al., 2020), so it has its scaling options, which are (0) un-scaled principal components, (1) principal components scaled to unit variance (divided by the square root of their eigenvalue) and (2) principal components multiplied by the square root of their eigenvalue. This is implemented for the boreal winter, spring, summer and fall, for four principal component as follows.

```
predictor_pcs = predictors.pcs(
    npcs=4,
    seasons=[[12, 1, 2], [3, 4, 5], [6, 7, 8], [9, 10, 11]],
    standardize=True,
    pcscaling=1
)
```

### 4.3 Analogs class

After establishing the predictor and determining its synoptic patterns via PCA, the next step is to search for a set of days with similar synoptic patterns for each day, known as the analog pool. To ensure the validity of the reconstructions, it is recommended to exclude at least the 5 previous and/or following days from the day to be reconstructed to avoid persistence effects. After determining the analog pool, the days without observations are reconstructed using one of the following similarity methods: (1) the 'closest' method, which selects the closest day in the space of the PCs, (2) the 'average' method, which calculates the weighted average of the N closest days, or (3) the 'quantile-map' method, which chooses the day that corresponds to the same quantile as the day to be reconstructed in another variable called the 'mapping variable'. These steps are implemented in the *Analog* class. This object is fed by the historical observations from the *Station* object and the PCs time series of the *Predictor* object. This object allows to select the number of analog days in the pool as *pool_size*, the number of days to exclude for the reconstruction validation, and whether to exclude the previous, posterior or both days, as *vw_size* and *vw_type* arguments. Additionally, it allows for the selection of the similarity method to be used as *method*. To use the quantile-mapping method a mapping variable is required. This variable must be a time series from the reanalysis dataset in the gridpoint of the station. The *Predictor* class can be used to obtain this information by setting the domain limits to the station's localization, which is saved in the *Station* object.

```
mapping_variable_files = rascal.utils.get_files(
    nwp_path="./data/reanalysis/",
    variables=["SURF_71.162", "SURF_72.162"],
```





```
dates=["1900", "1901", ..., "2010"],
             file_format=".grib"
             )

      mapping_variable = Predictor(
paths=mapping_variable_files,
             grouping="1D_mean",
             lat_min=station.latitude,
             lat_max=station.latitude,
             lon_min=station.longitude,
lon_max=station.longitude,
             mosaic=False
      )
      mapping_variable.module()

analogs = Analogs(
          pcs=predictor_pcs,
          observations=station_data,
          dates=[1900-01-01, 1900-01-02, ..., 2010-12-31]
      )

      reconstruction = analogs.reconstruct(
          pool_size=30,
          method="quantilemap",
          vw_size=10,
vw_type="centered",
          mapping_variable=mapping_variable
      )
```

### 4.4 RSkill class

To assess the quality of a reconstructed time series, it is necessary to clearly state its goal beforehand. RASCAL is designed
to produce daily reconstructions to calculate relevant indices, such as days above zero-degree isotherm or length of dry spells.
However, it is not necessary for the daily reconstructions to be completely in phase with the daily observations when the
objective is to evaluate these quantities at coarser temporal resolutions, such as monthly, seasonal, or annual. It is sufficient that
their behavior and statistical properties are well-reproduced at these coarser temporal resolutions. To evaluate the skill of the




reconstructions, RASCAL is equipped with an skill evaluation class called *RSkill*. This class contains functions to evaluate the behavior of the reconstructions and asses their added value compared to using the reanalysis data alone. The skill metrics and diagrams included are the following: Taylor diagrams, quantile-quantile diagrams, time series and annual cycles plots, Root Mean Squared Error (RMSE), Correlation cofficient ($R^2$), Mean Bias Error (MBE), MSE-based skill score (Where MSE is the Mean Squared Error), Heidke skill score (HSS) and Brier Score (BS).

The MSE-based skill score (Wilks, 2011) is given by:

$$SS_{MSE} = 1 - \frac{MSE}{MSE_r} \tag{1}$$

Where $MSE$ is the MSE of the reconstruction and $MSE_r$ the MSE of the reference model, in this case the reanalysis. Therefore, the $SS_{MSE}$ can be interpreted as the relative error reduction of the reconstruction compared to the reanalysis series.

The Heidke Skill Score (HSS) is implemented in order to assess the performance of the analog method in predicting days where the predictand is above or below a certain threshold compared to the reanalysis, based on a contingency table analysis. The HSS scores events based on their occurrence or absence and determines whether the performance of the tested model is superior to that of the reference model. The HSS is defined as:

$$HSS(r) = \frac{r - r_r}{1 - r_r} \tag{2}$$

Where $r$ is the proportion of correct forecast (true positive and true negative) of the reconstructed series, and $r_r$ the proportion of correct forecast of the reanalysis. The proportion of correct forecast is expressed as

$$r = \frac{a + d}{a + b + c + d} \tag{3}$$

Where $a$ is the number of times that en event is forecasted and observed (true positive), $b$ the number of times that is forecasted but not observed (false positive), $c$ the number of times that is observed but not forecasted (false negative) and $d$ the number of times that is neither forecasted nor observed (true negative).

This score condense the information whether the tested model performs better that the reference model with a number in the interval $(-\infty, 1]$. A model that perfectly reproduces the observations get a score of one, if it performs as well as the reference model it gets a score of zero, and if the model performs worse than the reference model it gets negative scores.

### 4.5 RASCAL implementation

Although RASCAL is designed as a Python library, the GitHub repository contains scripts that allow for perform reconstructions and skill evaluations without the need to write a script. The *multiple_runs_example.py* script demonstrates a workflow that reconstructs several stations and variables using different values for the parameters including analog pool size, number of days in the pondered average, and similarity method. There is also a Jupyter notebook available, named *RASCAL_evaluation.ipynb*, which evaluates the skill of the reconstructions for daily, monthly, and annual series.





# 5 Application examples

To test RASCAL performance, we tested the skill of the reconstructions of maximum and minimum temperature, and daily
precipitation at four different surface stations in the vicinity of the Central System of the Iberian Peninsula (Spain), as shown
in Figure 3. This mountain range is of vital importance as the main hydrological resource in central Spain and has been subject
of several studies by the authors in recent years (Durán et al., 2013, 2015; Durán and Barstad, 2018; González-Flórez et al.,
2022). The reader can refer to these previous works in order to understand the importance of having long time series in this
area.

## 5.1 Observational data

The surface stations used are sumarized in Table 1. All of the stations belong to the Spanish Meteorological Agency (AEMET,
http://www.aemet.es, accessed on 13 September 2023), which has conducted high-quality observations of precipitation and
temperature since 1893 for the case of 'Retiro' station located in Madrid. The 'Navacerrada' station is the highest one, sit-
uated at 1888 m.a.s.l. in the core of Sierra de Guadarrama, an area that has been kept almost unaltered since then. Stations
'Segovia' and 'Colmenar' are located on the northern and southern slopes, respectively, of this mountain massif (Fig. 3). These
set of surface meteorological stations were selected based on their long historical records, on the variety of their orography
and environments, as well as on the deep knowledge of this area due to previous research carried out by the authors of this
work. Furthermore, they spread across a wide range of altitudes. Of the four data sets, two are particularly long: Retiro and
Navacerrada.

Observations have been available at Navacerrada station since 1946. This station is located at 1888 m.a.s.l. and has one of
the longest meteorological records for studying mountain meteorology in the world. The whole region has remained practically
unchanged since then, making it a valuable resource. In contrast, the Retiro station is located in the heart of the city of Madrid.
Observations of this station have been available since 1948. The city has undergone significant growth, particularly since the
1960s.

This set of observations can serve as a suitable test bed for evaluating the strengths and weaknesses of RASCAL and its
working hypotheses.

## 5.2 Reanalysis Data

The reconstruction of the time series was performed using ECMWF reanalysis data. Specifically, ERA20C data for the temper-
ature and ERA20CM ensemble data for the precipitation (Poli et al., 2016) were used for the period from 1900 to 2010, with a
spatial resolution of 0.75°x0.75° and a temporal resolution of 3 hours.

Principal component analysis was conducted for each season (DJF, MAM, JJA, and SON) individually, using geopotential
height (GpH) data at 925 hPa as a temperature predictor and TCWVF as the precipitation predictor. The 'quantile-map' method
used the 2m temperature and TCWVF to search for analogs in the dataset as the mapping variables. The selection of these



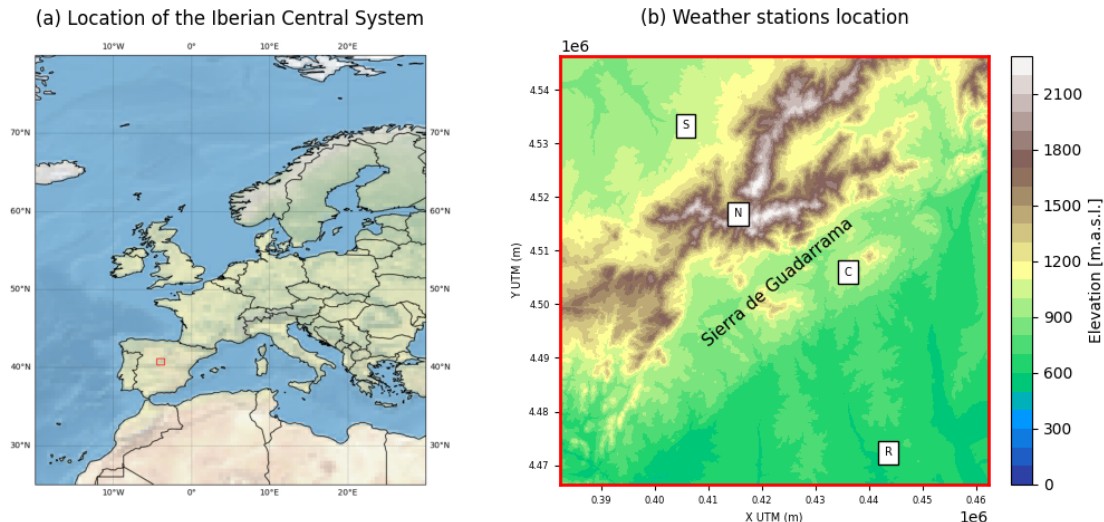

**Figure 3. (a)** Location of the Iberian Central System. **(b)** Location of sites used in this example of application, these being: S (Segovia), N (Navacerrada), C (Colmenar) and R (Retiro).

**Table 1.** Observational data used

| Station | Altitude | Coordinates | Variables | Period | Frequency |
|---------|----------|-------------|-----------|--------|-----------|
| Colmenar | 1004 m.a.s.l. | 40°41'46.000"N 3°45'54.000"W | Temperature | 1978-2023 | Daily |
| | | | Precipitation | 1978-2023 | Daily |
| Navacerrada | 1888 m.a.s.l. | 40°47'35.000"N 4°00'38.000"W | Temperature | 1946-2023 | Daily |
| | | | Precipitation | 1946-2023 | Daily |
| Retiro | 660 m.a.s.l. | 40°24'43.000"N 3°40'41.001"W | Temperature | 1893-2023 | Daily |
| | | | Precipitation | 1948-2023 | Daily |
| Segovia | 1000 m.a.s.l. | 40°56'48.012"N 6°6'56.998"W | Temperature | 1988-2023 | Daily |
| | | | Precipitation | 1948-2023 | Daily |

predictors was based on their previous use in identifying circulation weather types for precipitation and extreme snow events

in the study region, as reported by Durán et al. (2015) and González-Flórez et al. (2022).



## 5.3 Model evaluation

The reconstructions were performed for all stations, using all three similarity methods and varying values of pool size and number of days to average in the 'average' method, to account for the possible sensitivity of the results to these parameters. To evaluate the quality of a reconstruction, it is necessary to determine the similarity of the time series to the observations.
However, in climate studies it may be more relevant to consider the statistical characteristics of the series. Therefore, it may be more effective to evaluate the ability to reproduce daily distributions, seasonality, seasonal and interannual variability, and relevant indices, such as the number of days below zero degrees or days of precipitation above certain threshold. RASCAL evaluates the effectiveness of reconstructions for use in climate studies. It assesses the behavior of time series for maximum and minimum temperature, as well as precipitation, and their statistical properties. Additionally, it compares the performance
of using the analog method versus using reanalysis data to reproduce observations. RASCAL includes options to cross-validate when generating the reconstructions. For this application case, we excluded the five days before and after each day designated for reconstruction to avoid any persistence effects.

### 5.3.1 Time series skill

Taylor diagrams were implemented in RASCAL to assess the agreement between the reconstructed time series and the observed
data. As illustrated in Fig. 4, Fig. 5 and Fig. 6, these diagrams provide a visual representation of the analysis, displaying the standard deviation and correlation of each time series in comparison to the reference observations.

As depicted in Fig. 4 the precipitation reconstructions outperform the reanalysis precipitation in all the cases, both in total monthly and total yearly precipitation. Monthly reconstructions yield better results than the yearly series, with correlation coefficients ranging from 0.4 to 0.8, whereas the yearly series ranges from 0.2 to 0.7. However, for both cases, the reanalysis
only shows correlations of 0.4 at best, and negative correlations at worst (not visible in the diagram). The panels (c, f, i, l) display the yearly time series in water years (from October to September), comparing the observations with the reanalysis ensemble and the best reconstruction. The chosen reconstruction balances a good correlation coefficient and a standard deviation close to the observations. These panels demonstrate that not only are the correlation and standard deviation better than the reanalysis, but also that it corrects its biases. An example that illustrates this point is 'Navacerrada' (Fig. 4i), where the reanalysis dry bias
may be attributable to a smoothed reanalysis orography that hampers orographic precipitation, an important contributor to total precipitation in this region (Durán et al., 2017).

The reconstructions are somewhat sensitive to the pool size selection, but this sensitivity is not significant enough to be considered a critical determinant in the simulations. Therefore, adjusting this parameter can be useful for subtle modifications.

Additionally, the Taylor diagrams demonstrate how various scientific inquiries may require different similarity methods.
While 'average10' shows the strongest correlations, 'quantilemap100' exhibits standard deviations closest to the original series, resulting in more similar distributions. Therefore, the choice of reconstruction methods may depend on the specific goals that led to the reconstruction process.



**Figure 4.** Precipitation time series reconstruction skill for all the stations. The left panels **(a, d, g, j, m)** show the Taylor diagrams of the monthly total precipitation series. The central panels **(b, e, h, k, n)** show the Taylor diagrams of yearly total precipitation series. The right panels **(c, f, i, l, o)** show the time series of observations, reanalysis and the selected as best performing reconstruction. In the precipitation case the yearly series are based on water years, beginning in October and ending in September





Fig. 5 illustrates that the reconstructions of maximum temperatures yield better results than those of precipitation, with correlation coefficients above 0.93 for the monthly mean maximum temperature series and between 0.3 and 0.9 for the yearly mean maximum temperature series. The reanalysis exhibits a very similar behavior to the quantile-map reconstructions for this variable, but the latter consistently shows a slight improvement in correlation or standard deviation. The time series panels in Fig. 5c, f, i, l, show that although the behaviour of the reanalysis is very close to the observations, the reconstructions correct the bias for all the stations.

Maximum temperature reconstructions exhibit less agreement between different similarity methods, but demonstrate more consistent outcomes for different pool sizes, when compared to the precipitation reconstructions. In this case, the 'quantile-map' method is recommended for reconstruction, above 'closest' and 'average'.

In Fig. 6a, d, g, j, the monthly mean minimum temperatures exhibit a similar behavior to the maximum mean temperatures in Fig. 5a, d, g, j, with correlation coefficients above 0.9, and a slight improvement in the correlation and standard deviation compared to the reanalysis. The yearly mean series (Fig. 5b, e, h, k) also ahow moderate improvements compared to the reanalysis, with correlations ranging from 0.3 to 0.9. The 'quantile-map' method was found to be the most effective for reconstruction. The bias corrections are apparent in Fig. 6c, f, i, l.

It should be noted that the reconstruction of 'Retiro' in Fig. 6l, shows a peculiar behavior, overestimating the mean minimum temperatures before 1945, followed by an underestimation of the temperatures thereafter. This effect does not appear in the maximum temperature reconstructions in Fig. 5l, a possible hypothesis is that this is due to the progressive urbanization of Madrid, the city in whose core 'Retiro' is located. This urbanization leads to a change in the land use and an increase in the heat island effect, which affects mainly the increase in minimum temperatures (Yagüe et al., 1991). This induces a change in the relationship between the local scale and the synoptic scale, and therefore in the relationship between the predictor and the predictand, ultimately affecting the temperature trends.

### 5.3.2 Distributions

To evaluate the statistical properties of a reconstruction, we first examined the distributions of the daily time series in comparison to the observations. Fig. 7 displays the quantile-quantile plots of the daily time series for maximum and minimum temperature, as well as precipitation. These plots illustrate the values assigned to the same percentiles for the distributions of the reconstructed and observed time series. When the distributions are identical, the points align along a 45° line. The distribution of the observations is well represented by the 'closest' and 'quantile-map' methods, as shown in the first row of Fig. 7. However, the 'average' method affects the extreme values as expected, narrowing the distribution further as the pool size increases. The poor performance of the reanalysis in representing precipitation distributions is also evident, as it exhibits a skew towards lower values. All methods show a high alignment with the observed data regarding maximum and minimum temperatures, with a slight narrowing in the distribution for the 'average' methods. The impact of bias correction compared to the reanalysis is prominently noticeable in these variables as well.





**Figure 5.** Maximum temperature time series reconstruction skill for all the stations. The left panels **(a, d, g, j, m)** show the Taylor diagrams of the maximum temperature monthly mean series. The central panels **(b, e, h, k, n)** show the Taylor diagrams of yearly mean series. The right panels **(c, f, i, l, o)** show the time series of observations, reanalysis and the selected as best performing reconstruction.







**Figure 6.** Minimum temperature time series reconstruction skill for all the stations. The left panels **(a, d, g, j, m)** show the Taylor diagrams of the maximum temperature monthly series. The central panels **(b, e, h, k, n)** show the Taylor diagrams of yearly series. The right panels **(c, f, i, l, o)** show the time series of observations, reanalysis and the selected as best performing reconstruction.




**Figure 7.** Quantile-Quantile plot of the daily time series for all the stations (from left to right) including all the reconstructions and the reanalysis. The first row panels **(a, b, c, d, e)** are for the precipitation, the second row **(f, g, h, i, j)** for the maximum temperature, and the third row **(k, l, m, n, o)** for the minimum temperature.

### 5.3.3 Seasonality

Understanding the seasonality of meteorological variables is crucial for climate studies as it enables the identification of recurring patterns and trends throughout the year. As shown in Fig. 8 the seasonal cycles of total monthly precipitation and monthly standard deviation reveal that the reconstructions generally reflect the observations more accurately than the reanalysis. Notably, the quantile-map method shows better performance from January to June, while the closest and average methods work better from July to December. These differences between methods are more pronounced in MAM and SON, which are the months of highest variability (Fig. 8e, f, g, h). The standard deviation is well captured by the quantile-map method, with exceptions noted in February and November. 'Navacerrada' once again emerges as the station that benefited the most from the reconstructions. It exhibits the most similar precipitation cycles and standard deviation, outperforming the reanalysis.





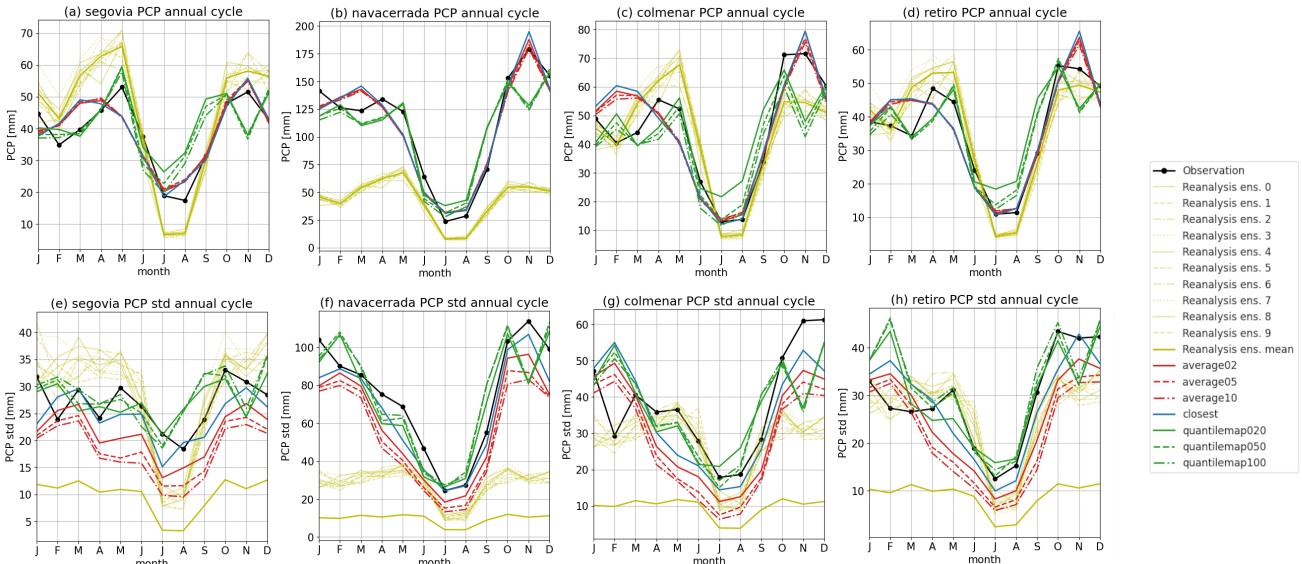

**Figure 8.** Annual cycle of monthly total precipitation for all the stations (left to right). The first row panels **(a, b, c, d, e)** show the cycle for the variable, and the second row **(f, g, h, i, j)** for its standard deviation.

Fig. 9 and Fig. 10 illustrate the annual cycles for maximum and minimum temperatures, respectively. The 'quantile-map' method outperforms the 'closest' and 'average' methods, as evidenced by the standard deviation, and is therefore recommended. These results demonstrate that RASCAL is more effective than the reference reanalysis in representing seasonality.

### 5.3.4 Daily Indices

In climate studies, it is common to employ indices that condense key climatic features of the study area. These indices are usually based on the comparison of a variable with a fixed threshold or a threshold based on some statistical property, such as a mean value or a percentile (Data, 2009). Consequently, when using climate indices, the focus of a study may not necessarily be on making the reconstructed time series closely resemble the observations, but rather on effectively reproducing these indices. Given that the indices are based on threshold crossings, a dataset characterized by significant biases may result in an misrepresentation of these indices. As demonstrated earlier, the station 'Navacerrada' stands out as the one most positively influenced by the reconstructions. This is due to the fact that the reanalysis provides a deficient representation of precipitation and temperature, mainly due to its pronounced warm and dry bias. Therefore, this station was chosen for the calculation of relevant indices for a mountainous region using the reconstructions, such as days with precipitation exceeding 1 mm (R1mm), icing days (IC, days of maximum temperature below $0°C$), and frost days (FD, days of minimum temperature below $0°C$).

Fig. 11 presents the seasonal cycles of the indices for both observations and the optimal reconstruction chosen in Section 5.3.1.. The aim is to ensure that the reconstructions accurately replicate the climatic characteristics associated with these indices without exhibiting any spurious behavior.





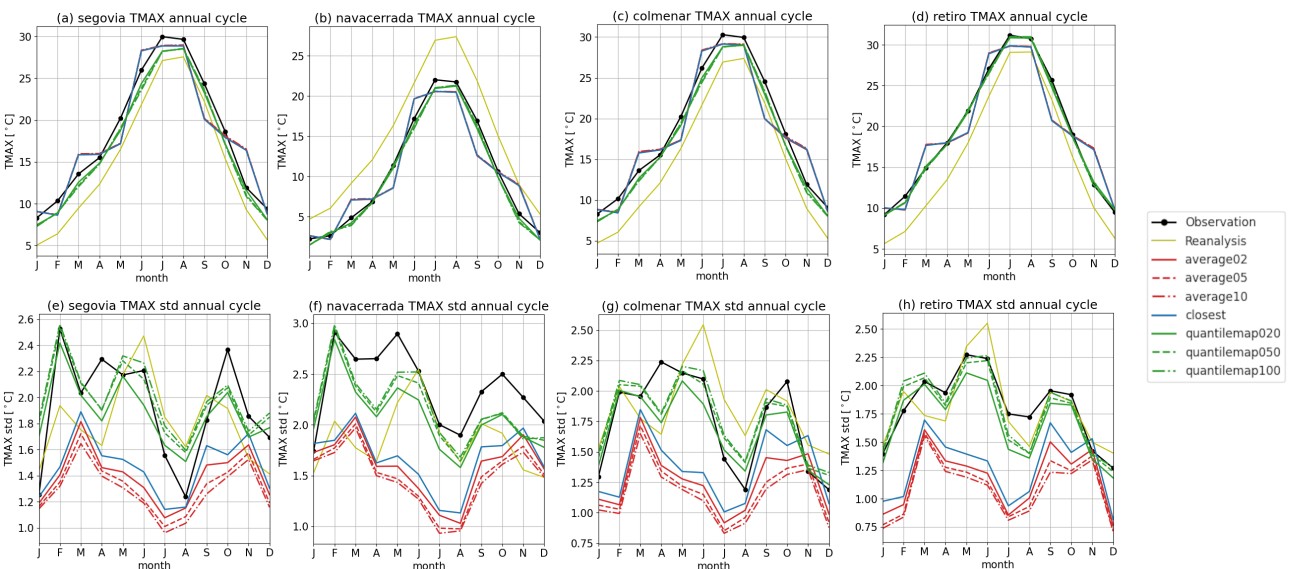

**Figure 9.** Annual cycle of monthly mean maximum temperature for all the stations (left to right). The first row panels **(a, b, c, d, e)** show the cycle for the variable, and the second row **(f, g, h, i, j)** for its standard deviation.

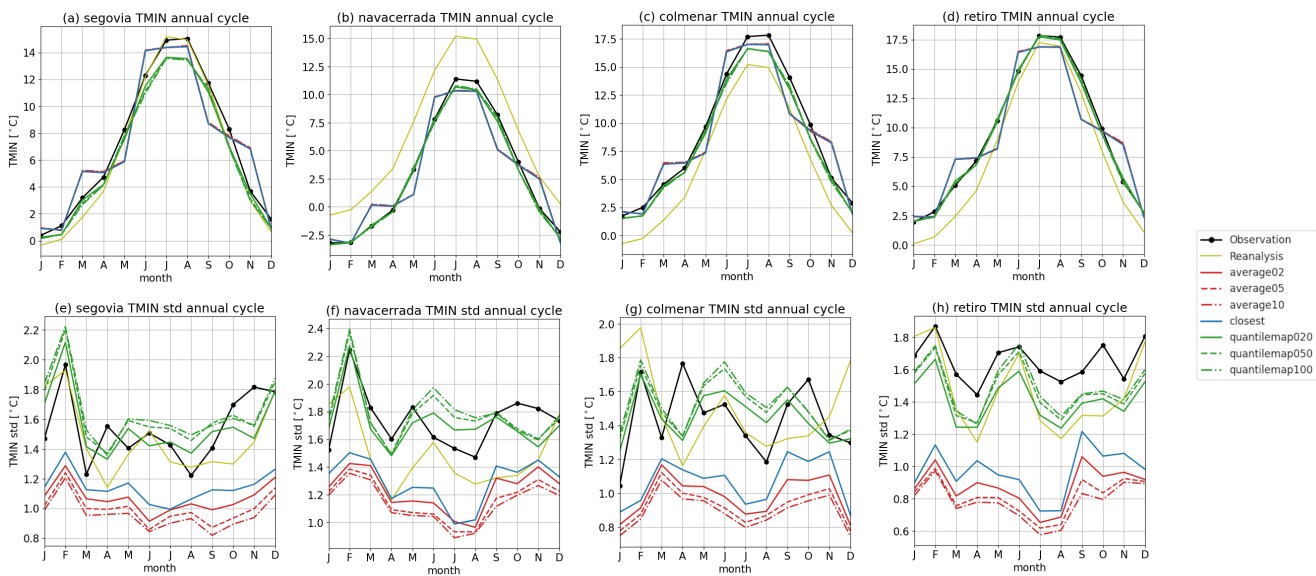

**Figure 10.** Annual cycle of monthly mean minimum temperature for all the stations (left to right). The first row panels **(a, b, c, d, e)** show the cycle for the variable, and the second row **(f, g, h, i, j)** for its standard deviation.



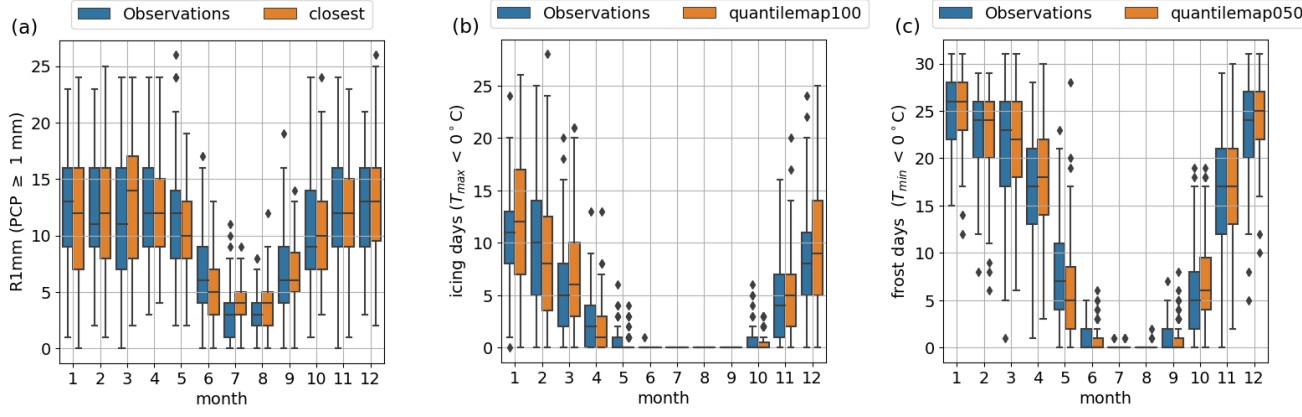

**Figure 11.** Seasonal cycle of the observations and the best reconstruction of climatological indices in Navacerrada, these being: **(a)** days of $PCP \geq 1mm$ (R1mm), **(b)** days of $T_{max} < 0^\circ C$ or icing days (ID) , and **(c)** days of $T_{min} < 0^\circ C$ or frost days (FD).

The R1mm index in Fig. 11a reveals highly similar distributions across all months, with only a slight overestimation of the median value noted in March, June, and July. Fig. 11b also demonstrates very good agreement in the ID index between the reconstruction and observed distributions and median values, although with slightly broader distributions during winter months. Finally, Fig. 11c reaffirms the substantial agreement between observations and distributions for the FD index, highlighting
RASCAL's capability to faithfully replicate the seasonal behavior inherent in these indices.

Upon examining the time series skill, Table 2 presents the values for Pearson correlation coefficient ($R^2$), Mean Bias Error (MBE), and Root Mean Square Error (RMSE) of the indices. The table highlights the commendable performance of the reconstructions in accurately reproducing these indices, as evidenced by high correlations, particularly for temperature-related indices. Furthermore, the MBE values are significantly low, measuring less than 0.34 days per month, and the RMSE values
remain below 3.55 days per month.

**Table 2.** Skill metrics for the best monthly reconstruction in Navacerrada of days of $PCP \geq 1mm$ (R1mm), icing days (ID), and frost days (FD).

| Index | $R^2$ | MBE [days month$^{-1}$] | RMSE [days month$^{-1}$] |
|-------|-------|--------------------------|---------------------------|
| R1mm  | 0.80  | 0.17                     | 3.55                      |
| ID    | 0.91  | 0.22                     | 2.3                       |
| FD    | 0.96  | 0.34                     | 3.01                      |





## 6 Conclusions

We have confirmed a decline in in situ observations is a noteworthy concern as it may result in the loss of crucial information in areas where local effects are relevant to their climatology. While the reanalysis provides a homogeneous and comprehensive dataset, its applicability to studying climatologies with highly localized effects, particularly in regions with intricate orography, has been called into question.

In order to mitigate this possible loss of meteorological information based on surface observations, RASCAL has been developed. This is an open-source Python tool designed to fill gaps in observational data, enabling climate studies in regions with limited long-term data. This tool has shown for the test sites particularly useful, specially in the mountain sites. It is expected to be also useful in other areas with important local effects or distinctive locations like river valleys, forests, caves, or canyons.

The package presented here utilizes an object-oriented programming (OOP) approach, treating weather stations, predictors, and reconstructions as objects with multiple functional attributes that encompass all necessary functionalities. This has allowed for the execution of all modeling steps with just a few lines of code.

The core methodology is based on linking large-scale circulation patterns with local atmospheric features. This linkage is established through the analog method and principal component analysis, and has been shown to be more effective than reanalysis in conveying climatic characteristics.

The package was evaluated at four stations in Spain, including three near a mountainous area in central Spain and one in a highly urbanized area. The results were compared to the products of the reanalysis ERA20C and ERA20CM. RASCAL outperformed the reanalysis in terms of $R^2$, standard deviation, and bias. This improvement was particularly noticeable in the reconstruction of monthly total precipitation, with correlation values reaching 0.8. The reconstructed maximum and minimum temperatures show a slight improvement over the reanalysis in terms of standard deviation and correlation, reaching very high values of correlation, achieving high correlation values of over 0.99 for both monthly maximum and minimum temperatures. Additionally, the biases present in the reanalysis are significantly corrected by the reconstructions. This is also evident when examining the distributions of daily data. RASCAL is proficient in generating series that closely resemble the observations, unlike the reanalysis, which exhibits skewness towards low precipitation and biases in maximum and minimum temperatures.

The various methods for selecting the best analog have exhibited diverse behaviors when examining the different characteristics of the series. Therefore, it is recommended not to designate a single method as the best possible, but to choose it based on the scientific objectives.

Seasonality also demonstrates a marked enhancement compared to the reanalysis. RASCAL produces reconstructions with an annual cycle closely resembling the observations. While the precipitation annual cycle exhibits some disparities during unstable months, such as November and March, the cycles of maximum and minimum temperatures are nearly identical to the observations in every month when using the 'quantile-map' method. This method better represents the monthly variability for both precipitation and temperatures.



In climate studies, the use of indices is a common practice to condense key climatic features of a study area. RASCAL
has demonstrated its capacity to reproduce well indices like days of precipitation above 1mm, icing days, and frost days, in a
station situated in the core of a mountain range. This achievement is particularly noteworthy given the difficult conditions posed
by the strong dry and warm biases of the reanalysis in this region, which would otherwise hinder the accurate computation
of these indices. The reconstructed data showcase high correlation coefficients with observations, ranging from 0.8 to 0.96.
Additionally, consistently low values of MBE and RMSE were observed. These outcomes highlight the significant potential
of RASCAL in facilitating climate studies in regions with complex climatic dynamics. These results confirm RASCAL's
effectiveness in capturing and reproducing important climatic features for reliable climate research, highlighting its potential
in regions with limited long-term weather data.

However, it is important to acknowledge instances where this methodology may have limitations. This approach requires of
long-time series, as it cannot reconstruct with data that has not been observed. Additionally, land use changes or urbanization
processes, can disrupt the intricate relationship between large and small scales, affecting the relationship between predictor
and predictand, and ultimately, the quality of the reconstruction.

The implementation of this package has yielded positive results, providing opportunities for conducting comprehensive
climate variability analyses within the study area. In a short time, it is expected to use RASCAL in the analysis of the climate
variability and climate change in the mountainous area of Central System (Spain). On the other hand, improvements to be
implemented in this methodology will be studied once it has been applied to different cases, scenarios and regions. Finally, it
will be analyzed if this package can be extended as a downscaling tool for short and medium term numerical forecast, as well
as for seasonal prediction and even climate change scenarios.

*Code availability.* RASCAL (version 1.0.0) source code is available in GitHub (https://github.com/alvaro-gc95/RASCAL) and Zenodo
(https://doi.org/10.5281/zenodo.10592595 (Gonzalez-Cervera, 2024)). The required dependencies, package usage and functionalities are
500 described in the GitHub documentation (last access: 28 March 2024). Additionally, a Jupyter Notebook is available to represent and validate
the reconstructions and assess their skill. To run this library, Python 3.7 or later versions are required. RASCAL is also installable via Python
package index (PyPI): https://pypi.org/project/rascal-ties/ (last access: 28 March 2024).

*Author contributions.* AGC and LD contributed to the conceptualization of the model and the writing of the paper. AGC contributed to
the RASCAL software coding, documentation, modeling, data analysis, and visualization. LD contributed to the supervision and funding
acquisition.

*Competing interests.* The contact author has declared that neither of the authors has any competing interests.



*Acknowledgements.* Partial funding comes from Ministerio de Ciencia e Innovación. Programa de doctorados industriales 2019, DIN2019-010482, Project FIRN (PID2022-140690OA-I00) Proyectos de Generación de Conocimiento 2022. Ministerio de Ciencia e Innovación, and interMET Sistemas y Redes SME. We would like to thank the Agencia Estatal de Meteorología (AEMET) for providing observational data. Special thanks to Navacerrada Observatory staff for their tireless and accurate work. Thanks to the European Centre for Medium-Range Weather Forecasts (ECMWF) for providing the ERA20C and ERA20CM Data. Thanks to Nuria for coming up with a cool name for the package. We would like to thank the referees for their thoughtful reviews that helped to make this a better paper.



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
