# Peer review of "RASCAL v1.0.0: An Open Source Tool for Climatological Time Series Reconstruction and Extension"

_EGUsphere, 2024_

## Author Comment (AC1)

**REPLY TO REFEREE 1**

**Summary**

Reanalysis products are globally gridded climate observations produced using numerical weather prediction models within a data assimilation framework (e.g., ERA5; Hersbach et al., 2020). These products often misrepresent climate variability and patterns in regions where localized conditions significantly influence climatology and where observational networks are sparse. However, reanalysis products effectively capture large-scale atmospheric dynamics (e.g., circulation patterns). This information can be combined with local observations to develop a transfer function, which can be used to downscale their data through statistical methods such as bias correction (e.g., ibicus; Spuler et al., 2024) and Perfect Prognosis (e.g., pyESD; Boateng & Mutz, 2023). Additionally, analogue models can be used to identify historical atmospheric patterns or weather states, which helps fill in missing observations or extend observations back in time.

The manuscript describes a Python package (RASCAL v1.0.0) that incorporates all the necessary steps for using analogue models and Principal Component Analysis (PCA) to combine reanalysis (ERA20C and ERA20M) and station observations (precipitation and temperature) to reconstruct and extend climatological time series (as illustrated for four stations in Spain). The manuscript is well-written and the authors have clearly presented their work. However, I have major comments that should be addressed before publication, along with some minor comments for further clarification or additional detail.

**General Comments**

**Introduction and motivation section**: The authors provide detailed information about the decline of climate observation networks and its consequences. However, the emphasis on these issues overshadows the main focus of the manuscript, which is gap-filling or reconstructing missing records. I encourage the authors to shorten the introduction and focus more on the methods used, as the paper does not address the global increase of observational networks.

You are right, thank you for your input. We agree that the line of thought of the introduction and motivation was not very clear. Following your remarks and in accordance with comments from the second referee we have rebuilt this section merging both sections trying to simplify the message and be more concise.

**Evaluation Metrics:** The manuscript lacks clarity on how the reconstruction was compared to actual observations. It would be beneficial if the authors elaborated on this. For instance, which period was used for training or identifying the analogue patterns, and which period was retained for independent performance assessment? Even if cross-validation was employed, a detailed explanation is necessary. Additionally, using reanalysis observations as a reference for evaluating reconstruction performance is questionable. Station-based reconstructions, being trained specifically for a station, will naturally outperform reanalysis. The critical aspect is the prediction error on the daily scale between observations and reconstructions for the period not included in model training. The authors should consider

splitting the data into training and testing sets or randomly excluding parts of the time series for evaluation.

Thank you for your feedback. We acknowledge that the manuscript lacked clarity on how the validation and comparison with observations were conducted. To address this, we have made the following revisions and clarifications:

For each day to be reconstructed, we excluded the N closest days in time to avoid including the most probable analogs, which are often due to the persistence of atmospheric patterns. In the initial version of the manuscript, for the calcutaions we excluded a window of 10 days around each target day (5 days before and 5 days after). This exclusion ensures that the large-scale pattern and its associated local weather on those days are not part of the analog pool used for reconstruction. After careful consideration and following your remarks, we determined that the original window was probaly too narrow and we have recalculated the series using a centered window of 60 days.

We have added a detailed explanation of this methodology in the revised manuscript (see lines 309 to 314).

**Model Development:** The package documentation needs improvement. More detailed instructions on using the package and its functionalities should be provided. Although the documentation will evolve as user numbers grow, the current state is insufficient for users to get started easily. If this is done properly, the code snippets in the manuscript can be removed and referred to the documentation website.

You are right. Thank you for your feedback. We deleted the code snippets as we expanded and completed the documentation. Including clearer tutorials of how to install the model (https://rascalv100.readthedocs.io/en/latest/installation.html) and how to use it (https://rascalv100.readthedocs.io/en/latest/begginer.html) with example data uploaded in Zenodo (https://zenodo.org/records/12626857), and all its modules and functionalities (https://rascalv100.readthedocs.io/en/latest/code.html)

**Package Installation:** I encountered issues while installing the package from both GitHub and PyPi. The PyPI upload lacked the necessary modules, causing import errors post-installation. The GitHub repository also lacked a setup configuration file, complicating installation after cloning. I urge the authors to test the installation in an independent environment to ensure functionality and highlight this in the installation section of the documentation.

Thank you for your input. We included a setup.py file in the github and we uploaded a new stable version to PyPI, and tested it on different computers with independent environments, working in all of them when the instructions in the documentation are followed.

**Testing the Package:** The authors should consider uploading synthetic datasets or the actual datasets used in the illustrative study. This will enable users to test the package before adapting it to their needs. Current software design decisions seem to rely heavily on the specific data used, which may hinder easy testing by new users. I believe reducing the

size of the predictors (reanalysis data) to only the area used in the study or even the PCA time series will not be that huge to upload on Zenodo as supporting material for the manuscript.

You are right, due to restrictions with the public distribution of AEMETs data we could not upload data used for the paper. However, we uploaded a reanalysis dataset to Zenodo (https://zenodo.org/records/12626857) and observations from our own independent nearby station to github (https://github.com/alvaro-gc95/RASCAL/tree/master/data/observations/St03).
The Jupyter notebook RASCAL_evaluation.ipynb was also modified to evaluate the example series and facilitate users to get used to the model, using the example data.

These new datasets are not restricted, so we hope that can be useful for the users for testing the package.

By addressing these comments, the manuscript can be significantly improved, providing a more robust and user-friendly tool for climate observation gap filling and extension

**Specific comments**

L3 *"Reanalyses face challenges in examining.."* change examining to representing

Done.

L4-5: add a sentence about how empirical downscaling resolves the issue of reanalysis mentioned

We have changed the abstract modifying the following sentence:

"Empirical downscaling methods offer a cost-effective and easier to implement in new areas alternative to dynamic downscaling methods. "

To:

 "Empirical downscaling methods offer a cost-effective and easier to implement **alternative to dynamic downscaling methods and can partially overcome the aforementioned limitations of reanalyses taking into account the local effects through statistical relationships.**"

L6: "*designed to address gaps in observational climate data..*" modify the sentence to highlight how it addresses gaps (designed to fill gaps?)

We changed sentence:

"This article introduces RASCAL, an open-source Python tool designed to address gaps in observational climate data, …"

To:

"This article introduces RASCAL, an open-source Python tool designed t**o extend and fill gaps in observational data.**"

L9: "*outperforming reanalysis in conveying climatic characteristics.*" more details is missing before this statement. Maybe only mention this when you've introduced the basis of the evaluation

The sentence " …outperforming reanalysis in conveying climatic characteristics. " has been erased.

L18-19: *"However, there are challenges to consider, such as the requirement for long-term data series and susceptibility to disruptions caused by changes in land use or urbanization processes.."* the author mentioned that the package is useful for addressing gaps in observational data with limited long-term data. But is mentioned here that a long-term data series is required. Please clarify. Also, rephrase susceptibility to disruption to something specific.

We agree it sounds contradictory, but we mean "long enough" time series, as the goal is to extend these series, that might be long enough to capture the  climatic characteristics but not long enough for robust climate studies. This sentence change from:

"However, there are challenges to consider, such as the requirement for long-term data series and susceptibility to disruptions caused by changes in land use or urbanization processes.."

*to:*

"However, **as with any other method based on empirical training, this method requires the availability of sufficiently long-term data series. Furthermore,** it is susceptible to disruption caused by changes in land use or urbanization processes that might compromise the homogeneity of the training data. "

L52: *"implicitly account for all the involved physics through complex mathematical.."* delete all here since such statement is overstretching

You are right. We deleted the sentence.

L66: "*satellite measurements are crucial for assessing the Earth's atmospheric conditions and perform the numerical weather prediction..*" you meant "..for monitoring Earth's atmospheric conditions and evaluating the performance of numerical weather predictions"

As mentioned before, we agree that the line of thought of the introduction and motivation was not very clear. Following your remarks and in accordance with comments from the second referee we have rebuilt this section merging both sections trying to simplify the message and be more concise. So these sentences are no longer in the manuscript.

L67: *"thanks to satellites, they may be behind the gradual decrease in the number of operational surface meteorological stations around the world.."* I couldn't understand the point raised here. Maybe modify the sentence and rephrase *"thanks to satellites"*

This sentence is no longer in the manuscript. This sentence might explain the idea in a better way:

 "One potential explanation for this decline in the number of surface stations is the advent of satellites as a novel method for observing the weather and climate. Following the launch of the world's first weather satellite, TIROS-1, in 1960, satellite weather observations became prevalent and began to offer a number of advantages over on-site weather observations, as cited in Purdom (1996). "

L70-71: I don't understand the point raised here too since reanalysis data also relied on weather stations. Please kindly clarify.

You are right, reanalyses use data from many sources, including surface weather stations. However, these stations are usually intended to be representative of a larger area in order to achieve a large spatial representativeness, similar to the spatial resolution of the reanalysis. We are not concerned here with the number of such stations.  Our hypothesis here is that there may be climate analyses performed with reanalysis data that would have required in-situ observations.

L86-87: I don't understand what the authors meant by may be behind the steady and heterogeneous decrease observed… please clarify.

Following your suggestions, we have made significant changes to the introduction and motivation to make these points clear.

L92: *"are the result.."* change to as a result of

The sentence was corrected.

L98-100: The author should be specific about the categories of downscaling. This should be Dynamical downscaling (e.g., RCMs) and Empirical Statistical Downscaling(ESD). ESDs are grouped into Model Output Statistics, Perfect Prognosis and Weather generators (in which analogue models are used).

You are right, changes were made accordingly.

L109: *"climate change scenarios.."* you mean climate change information

This sentences was erased with the new version of the manuscript.

L112: The authors should clarify what is meant by the regionalization method

"Regionalization" was changed to "downscaling".

L139: The resulting PCs from the EOFs are predictors or predictand?

You are right, this was a typo. PCs are predictands. It was changed in the manuscript.

L281: So if the objective is to evaluate low temporal resolutions like monthly, why not train the model directly with monthly data? The authors should present more evidence as to why it is accepted to train the model on high resolution but evaluated on low resolution

In order to calculate indices that have daily resolution, such as frost days or wet days, it is necessary to work at daily resolution for downscaling, although the results are later upscaled to monthly resolution for validation.

L316: The authors should clarify what they meant by the main hydrological resource

The sentence:

"This mountain range is of vital importance as the main hydrological resource in central Spain and has been subject of several studies by the authors in recent years…"

has been changed to:

"This mountain range is of vital importance as it is the main **contributor to the hydrological resources of central Spain, due to the high levels of rainfall and snow runoff in spring.** This area has been subject of several studies by the authors in recent years…"

L331-332: The authors should provide references to support the statement about the longest records

You are right, although we are aware of the great value of this observatory for the study of mountain weather in the Mediterranean and Western Europe, we agree that the sentence "This station has one of the longest meteorological records for the study of mountain meteorology in the world" may be too categorical. As it does not add much to the main point of the research, it has been deleted.

L341-342: The authors should provide more details on why these predictors were selected and also test the sensitivity of the reconstruction to other predictors. Even though the authors mention that details are presented in other studies, a summary of the reason here would be useful to readers

Other predictors have been used during development with good results. Different geopotential levels have been used with similar results. Since the purpose of this work is not to obtain the best results for the time series used, but to show the applicability and skill of the

method, we leave this process to the user. Adding such an analysis would probably result in a too long paper.

L369: You mean Figure 4f

You are right. It is corrected now.

L372: Where is it shown that the reconstruction is sensitive to the pool size selection?

It's true that this is not clear in the text. The sentence:

"The reconstructions are somewhat sensitive to the pool size selection, but this sensitivity is not significant enough to be considered a critical determinant in the simulations."

has been changed to:

"**As evidenced in the difference in the location of points of the same color in the Taylor diagrams, the reconstructions are somewhat sensitive to the pool size selection, as one method with different pool sizes can produce series with different correlations to the observations and standard deviations. However,** this sensitivity is not significant enough to be considered a critical determinant in the simulations."

L374: What is scientific inquires?

You are right. That sentence is not clear. The sentence:

"Additionally, the Taylor diagrams demonstrate how various scientific inquiries may require different similarity methods."

has been changed to:

"Additionally, the Taylor diagrams demonstrate how different **scientific questions** may require different similarity methods."

**References**

Hersbach, H., Bell, B., Berrisford, P., Hirahara, S., Horányi, A., Muñoz-Sabater, J., et al. (2020). The ERA5 global reanalysis. *Quarterly Journal of the Royal Meteorological Society*, *146*(730), 1999–2049. https://doi.org/10.1002/qj.3803

Spuler, F. R., Wessel, J. B., Comyn-Platt, E., Varndell, J., & Cagnazzo, C. (2024). ibicus: a new open-source Python package and comprehensive interface for statistical bias adjustment and evaluation in climate modelling (v1.0.1). *Geoscientific Model Development*, *17*(3), 1249–1269. https://doi.org/10.5194/gmd-17-1249-2024

Boateng, D., & Mutz, S. G. (2023). pyESDv1.0.1: An open-source Python framework for empirical-statistical downscaling of climate information. *Geoscientific Model Development Discussions*, 1–58. https://doi.org/10.5194/gmd-2023-67

---

## Author Comment (AC2)

**REPLY TO REFEREE 2**

**Overview**

This study addresses the decline in in-situ observations in climate reanalysis, aiming to enhance localized meteorological information by leveraging its connection to large-scale patterns via the analog method. The authors have created an open-source, object-oriented Python package to streamline the workflow. Validation was performed against widely used reanalysis products at four stations in Spain, encompassing three mountainous areas and one urbanized area. The method demonstrated improvements in key statistical measures.

Overall, the manuscript is of high quality, featuring well-structured text and informative figures. The reasoning process is scientifically sound, and the analysis is precise. The method has been shown to work as expected. I have only a few minor comments listed below. Once these are addressed, I recommend the work be accepted for publication.

**Specific comments**

- The writing of the last two paragraphs of the Introduction section is a bit disconnected. What's missing is the description of how this study will fill the mentioned gap.

  We agree that the line of thought of the introduction and motivation was not very clear. Following your relevant remarks and in accordance with comments from another referee we have rebuilt this section merging both sections trying to simplify the message and be more concise. We hope it is more clear now.

- The analog method: it would be more precise if the authors could summarize the method in mathematical equations in addition to the text description.

  You are right. We added a mathematical description of the method in section 2 "Method"

- Font size in Figs. 4-7: the font size of the legends is a bit small. Please adjust to make them more readable.

  You are right, we changed the font to a bigger one.

- The legends in Figs. 4-7, etc.: it's not necessary to label every ensemble member. There's no way that the readers can really distinguish them. Using one line/marker style with one label saying "Reanalysis ensemble members" is just enough.

  Thank you for your feedback, we changed it now.

- The Jupyter notebook in the Github repository (https://github.com/alvaro-gc95/RASCAL/blob/master/RASCAL_evaluation.ipynb) has a cell with errors in section "1.9) Yearly Taylor diagram", with the later sections not executed as a showcase. It would be more informative if the author can run them through for the potential users. After all, Jupyter notebook is not only about sharing the code but also about presenting the results.

  As the Jupyter notebook is designed to serve as an example of code and a tool for the validation of user-generated reconstructions, we have modified it to run the validation of a reconstruction of an example station, given that we are unable to disseminate the original AEMET observation data. The new Jupyter notebook is fully functional for new users and all the cells are run with the new example data, so now it does not present the data used in the original manuscript.

- It seems that the documentation website (https://rascalv100.readthedocs.io/en/latest/index.html) is still under construction. As a software endeavor, the documentation should be in a finished status before the manuscript gets published.

  You are right. The documentation website has been expanded, adding a more detailed description of the installation of the package, and tutorials with code snippets to facilitate its usage by new users.

---

## Author Comment (AC3)

**REPLY TO REFEREE 3**

This paper describes the structure and performance of an open tool developed by the authors to complement and extend climatological time series based on statistical downscaling of reanalysis/GCM data. It could contribute substantially to climatological studies especially on statistical properties of climatic variables at stations subject to localized effects. My comments are as follows.

1. Since selection of appropriate predictors is crucial to acquire the best estimation, it is desirable to state more specific and systematic methodology to select predictors from numerous meteorological variables in the reanalysis/GCM dataset at lines 186-188 and lines 341-343.

The number and variety of scientific questions that can be addressed through downscaling is considerable. This work aims to facilitate the utilization of analogue downscaling by potential users, although the systematic selection of predictors may prove challenging. As a general principle, the predictor should be one of the main forcing fields of the local variable; however, we refrain from providing guidance to avoid influencing the research process.

Anyway, following your suggestion we have modified the sentence:  "After selecting a predictor variable that is expected to have a strong relationship to the predictand variable, it is necessary to choose a predictor field domain that can identify relevant synoptic patterns for the study area."

To:

"After selecting a predictor variable that is expected to have a strong relationship to the predictand variable, **for instance its main large-scale forcing field, and that is relevant to the proposed scientific question,** it is necessary to choose a predictor field domain that can identify relevant synoptic patterns for the study area."

2. Since RASCAL is a downscaling tool of reanalysis/GCM data, I think its strength comparing to existing downscaling methods should be clarified rather than describing improvement from the reanalysis data in detail (in conclusions for example).

We think the main advantage of using RASCAL is that it is fully documented, tested and publicly available It is therefore possible that this will facilitate the learning curve of future users or encourage the utilization of this technique among those with less developed programming abilities. The following paragraph sustents this idea

"This work introduces and explains RASCAL v1.0 (Reconstruction by AnalogS of ClimatologicAL time series), an open-source tool for climatological time series reconstruction and extension using ESD. The primary goal of RASCAL is to promote and accelerate rigorous climate research in regions where surface meteorological observations are insufficient for climate analysis and where relevant regional and local meteorological processes can only be captured through in-situ observations. RASCAL could prove highly

beneficial for mountain climate research and other areas with unique microclimates, such as river valleys, forests, caves, or canyons."

However, to convey its strength compared to other methods we included the following sentence in the conclusions, in lines 419-421:

"…a faster and less computationally expensive alternative to dynamical downscaling methods, and an easier to interpret method than machine learning statistical downscaling methods."

3. To evaluate the real performance of the estimation model, observed data used for model validation should be excluded from model calibration. (For example, by splitting the observed period into calibration and validation periods)

This was already done, but we acknowledge that the manuscript lacked clarity on how the validation and comparison with observations were conducted. To address this, we have made the following revisions and clarifications:

For each day to be reconstructed, we excluded the N closest days in time to avoid including the most probable analogs, which are often due to the persistence of atmospheric patterns. In the initial version of the manuscript, we excluded a window of 10 days around each target day (5 days before and 5 days after). This exclusion ensures that the large-scale pattern and its associated local weather on those days are not part of the analog pool used for reconstruction. After careful consideration and considering your feedback, we determined that the original window was too narrow and recalculated the series using a centered window of 60 days.

We have added a detailed explanation of this methodology in the revised manuscript (see lines 309-314).

4. There are many options (closest, average, quantile) to adjust estimation to observation data. In such case, I think robustness of the selected option should be tested to make sure if it is valid only for the calibration data. (For example, again, by splitting the observed period into two or several periods and comparing the selected option)

As stated in the previous answer, the splitting was already done, but we detailed it better in the new version of the manuscript.

5. For temperature reanalysis data for the four stations in chapter 5, are they corrected according to difference of elevation? It seems simple elevation correction using typical adiabatic lapse rate gives quite good estimation. The reconstructed data should be compared to the corrected reanalysis data.

We made the elevation correction to the reanalysis data in the new graphs using the environmental lapse rate (-6.5ºC/Km). Although the bias of the temperature was corrected and gives better estimations than before, there is still some bias in some of the series. Also, as the taylor diagrams shows (Figures 4-6), the correlation and standard deviation are still better represented by the reconstructions, the quantile-quantile plots (Figure 7) show better

representation of the observation distributions, and the seasonal cycle (Figures 8-10) is also closer to the observations.

We specified this correction to the reanalysis data in lines 306-307:

"To compare its performance against the reanalysis, the temperature data of the reanalysis was corrected with the elevation using the environmental lapse rate (−6.5∘C/Km)"

The followings are minor corrections for specific words or phrases.

6. L16-17: "thus…biases" is repeated.

The sentence was corrected. This sentence was erased.

7. L56-57: "From a few hundreds of…to several thousand" According to Fig.1a, several thousand (around 7,000) at the end of the nineteenth century and several tens of thousand (around 30,000) at the end of the twentieth century.

The sentence was corrected.

8. L129: Title of Section 3.1 appears here, but Section 3.2 doesn't exist.

Solved. All content has been consolidated into a single section, designated as section 2, as previous section 2 has been incorporated into section 1.

9. L218: "In this method is possible" > "In this method it is possible"?

The sentence was corrected.

10. L219: "The the number of" > "The number of"?

The sentence was corrected.

11. L321: "sumarized" > "summarized"

The sentence was corrected.

12. L322-323 and L333: "observations of precipitation and temperature since 1893" and "Observations…available since 1948" are confusing as explanations of the data period of Retino station.

You are right, we changed the sentence from:

"Observations have been available at Navacerrada station since 1946. This station is located at 1888 m.a.s.l. and has one of the longest meteorological records for studying mountain meteorology in the world. The whole region has remained practically unchanged since then, making it a valuable resource. In contrast, the Retiro station is located in the heart of the city of Madrid. Observations of this station have been available since 1948. The city has undergone significant growth, particularly since the 1960s."

To:

"Observations have been available at Navacerrada station since 1946. The whole region has remained practically unchanged since then, making it a valuable resource. In contrast, the Retiro station is located in the heart of the city of Madrid, which has undergone significant growth, particularly since the 1960s."

13. L389: "ahow" > "show"

The sentence was corrected.

14. Table 1: It is better to change the order to S, N, C, R according to the order in Figs. 4-10.

The sentence was corrected.

15. Fig.4, Fig.5, Fig.6 captions: "a, d, g, j, m" > "a, d, g, j", "b, e, h, k, n" > "b, e, h, k", "c, f, i, l, o" > "c, f, i, l"

The sentence was corrected.

16. Fig.7, Fig.8, Fig.9, Fig.10 captions: "a, b, c, d, e" > "a, b, c, d", "f, g, h, l, j" > "e, f, g, h", "k, l, m, n, o" > "I, j, k, l"

The sentence was corrected.